# The Influence of Paradoxical Leadership on Adaptive Performance of New-Generation Employees in the Post-Pandemic Era: The Role of Harmonious Work Passion and Core Self-Evaluation

Naiwen Li and Mingming Ding *

School of Business and Management, Liaoning Technical University, Huludao 125100, China
* Correspondence: lgddmm@outlook.com

**Abstract:** The post-pandemic era is full of instability and uncertainty, which brings new challenges and opportunities to the development of organization. As a sustainable feature of enterprises, improving employees' adaptive performance levels is a necessary condition for enterprises to achieve the sustainable development goal. This study is based on self-determination theory, which focuses on new-generation employees as the key force of enterprise and incorporates harmonious work passion and core self-evaluation into the research framework to explore the influence of paradoxical leadership on adaptive performance. The survey data obtained from 519 new-generation employees shows that: paradoxical leadership is significantly and positively correlated with adaptive performance of new-generation employees; the relationship between paradoxical leadership and adaptive performance is partially mediated by harmonious work passion; core self-evaluation positively adjusts the relationship between paradoxical leadership and harmonious work passion. In addition, core self-evaluation also regulates the intermediary role of harmonious work passion—that is to say, the higher core self-evaluation of new-generation employees is, the stronger the intermediary role of harmonious work passion. The research results explain the connotation of how paradoxical leadership improves adaptive performance of new-generation employees, reveal the medium of the relationship between the two, and find both the role boundary of paradoxical leadership and the strategy to improve adaptive performance.

**Keywords:** paradoxical leadership; harmonious work passion; core self-evaluation; adaptive performance; new-generation employees





## 1. Introduction

In the post-epidemic era, the dramatically changing environment and fierce competition have presented even higher requirements for the adaptability and agility of enterprises [1]. Any duly responses to such changes are the everlasting development vitalities for any of the enterprises to win in the future, while the employees who effectively adapt themselves in the rapidly changing work situation are to be the solid foundation for the success of the enterprise [2]. Abundant knowledge resources and flexible adaptability are the core production elements of an enterprise [3]. As the backbone of enterprise, new-generation employees are the key resource in the rapid response and adjustment strategy of enterprise, and their professional quality and active thinking are the vital nutrients in the enterprise transformation and change. However, the ideology and behavior habits of this group are different from the rest of the intergenerational personnel because of their special social background and training environment [4]. How to effectively tap the expertise of new-generation employees, fully awaken their work potential in the changing situations, and effectively stimulate them to manifest more adaptive behavior in the enterprise bottleneck breakthrough has become a crucial issue, and should be well thought about by all enterprises.

The employees' adaptive performance is the adaptive behavior of employees facing the dynamic environment and changing tasks [5]. It is a beneficial supplement to the traditional task and peripheral performances, and plays an important role in promoting the changes and developments of organizations, teams, and individuals [3,6]. In organizational performance management, the adaptive performance problems of new-generation employees have attracted much social attention. Conversely, the adaptive performance is full of contradictions and tension in the completion process [7]. Employees are required to have certain knowledge literacy, acuity and adaptability. However, the new-generation employees not only have strong learning skills and adaptability, but are often also headstrong in their hopes to get the support of the organization and play independently [8,9]. In contrast, new-generation employees who have initially had the potential to implement adaptive behavior are precious resources in the enterprise transformation. But how can leaders, as organizational spokesmen, activate the adaptive behavior toward their employees of the new generation? Previous scholars have studied authentic leadership [10], servant leadership [10,11], self-leadership [12,13] and shared leadership [14] mechanisms by which equal types of leadership influence adaptive performance. However, the leadership behaviors that scholars have paid attention to in the past are based on the concepts put forward in the western cultural background and transferred to the Chinese situation, and the compatibility to solve the adaptive performance of new-generation employees to Chinese-specific local situations needs to be improved. In addition, these "pure and not miscellaneous" leadership methods are difficult to flexibly respond to the complex problems that are faced by the current enterprises in the post-epidemic era from multiple perspectives, nor do they meet the development needs of the new-generation employees. Therefore, if leaders are still facing new-generation employees in the same way as mentioned above, they might not achieve the desired adaptive performance improvement effect. Unlike previous studies, in this paper, we are focused on the research of paradoxical leadership, and we closely combine the native dual properties. First of all, the paradoxical leadership is a composite emerging leadership style with local characteristics based on the Chinese philosophy of Yin/Yang [15], with strong location and era characteristics, which can better interpret the influence mechanism of the leadership style on adaptive performance of new-generation employees in the Chinese situation. Secondly, in face of the dynamic and changeable enterprise environment, compared with the previous single leadership style, the paradoxical leadership is good at using contradictory integration thinking [15,16], properly handling the internal and external conflicts and problems of enterprises in complex situations, purposefully responding to the changes from all parties, and adopting a two-pronged approach to achieve strategic agility [17]. Likewise, the paradoxical leadership dialectical and unified thinking mode and the flexible response behavior can more effectively attract new-generation employees to learn and imitate [18], which might be helpful to improve adaptive performance of new-generation employees, thereby further enhancing the organization. In addition, paradoxical leadership will give their employees high requirements and multiple resources in their work [15], which is in line with the concerns of new-generation employees. Conversely, the high requirements are particularly important to stimulate new-generation employees to adapt themselves to the internal and external environment and task changes. This is because the downgrade of the matching work may make new-generation employees feel very much depressed due to the unpaid ambition; however, the high-expectation work can provide new-generation employees a full display platform, thereby greatly arousing their fighting spirit [19]. Conversely, the new-generation employees who can receive these leadership supports and assistance resources will often adjust their behavior more quickly to flexibly respond to changes in the organization [7,20] and their adaptive performances will also be improved. Therefore, this study will thoroughly examine the relationship between paradoxical leadership and adaptive performance of new-generation employees.

The existing research is mainly derived from the sense of work prosperity [21], self-efficacy [22], fair perception [23] and the psychological security [24] from other individual

cognitive factors to study the influence mechanism of paradoxical leadership on employee behavior or performance. Few researchers have discussed the influence of paradoxical leadership on employees' adaptive performance from the perspective of individual cognition and emotional integration. Nevertheless, harmonious work passion should be a positive state produced by individuals in their work, which is an integrated construct integrating cognition and emotion. The individuals with harmonious work passion are often accompanied by their strong work motivation [25,26]. Jundt [7] stated that the improvement of employees' adaptive performance is often the result of the integration of various factors such as individual cognition, emotion and motivation. Therefore, from the perspective of integration, it will be very much valuable to introduce the harmonious work passion of cognition, emotion and motivation to explore the influence of leadership on employees' adaptive performance. According to the theory of self-determination, paradoxical leadership can not only be conducive to creating a healthy and warm organization atmosphere in which employees realize self-worth, but can also give their employees moderate autonomy to play their own advantages, which, to a certain extent, can meet the requirements of new-generation employees' relations and competent and independent psychological demands, which can in turn trigger the employees' harmonious work passion with their provision of the energy and support [27]. The internal motivation can be stimulated by harmonious work passion, which is always regarded as the driving force of the employees' good work performance. This improvement may promote individuals with stronger senses of responsibility and overall concept in the face of complex and changeable situations. These employees will always consequently manifest much better performances such as concentration and executive control [28], which can purposefully achieve the goals to improve the employees' adaptive performance. This study, on the basis of the research mentioned above, predicts that harmonious work passion may have an intermediary role between paradoxical leadership and adaptive performance of new-generation employees. The self-determination theory suggests that the influence of the degree of external environment support on individuals varies depending on the individual's own traits [29]. It can also be said that individual passion for harmonious work will depend, to a certain extent, on their self-characteristics while the individuals with positive self-cognition will tend to show higher passion for harmonious work. In addition, the employees' personal traits may interfere with the impact of the leadership style upon the employees' cognition, emotion, behavior or performance. Even if when they are facing the same leadership style, employees with different personal traits may react differently [30]. The employees' core self-evaluation is often considered to be a deep personal trait, which reflects the employees' evaluation and cognition of their own strength and life significance [31], which plays an important role between leadership behavior and employee response [32]. Therefore, this study speculated that core self-evaluation, as an important trait of an individual, may have a contingency influence on the action process of paradoxical leadership.

Previous studies on adaptive performance analyzed all employees as a whole, but failed to draw attention to the limitations of a single leadership style to improve adaptive performance of new-generation employees. The existing studies are short of the mechanism of paradoxical leadership affecting adaptive performance of new-generation employees. Considering the gap within existing research, this study explores the influence of paradoxical leadership on the adaptive performance of new-generation employees in the post-epidemic era, and the intermediary and moderating role of harmonious work passion and core self-evaluation in the above relationship. Specifically, first of all, this study discusses why paradoxical leadership can improve adaptive performance of new-generation employees. Secondly, it discusses how paradoxical leadership can promote the adaptive performance of new-generation employees through harmonious work passion. Finally, it discusses when core self-evaluation can moderate the relationship among paradoxical leadership, harmonious work passion and adaptive performance of new-generation employees.

In summary, the following four main contributions will be made in this study: First, in the existing literature studies, employee adaptive performance has been always studied

as a package of overall research, but the uniqueness of new-generation employees as the subject of the enterprise has not always been taken into account, and this the research gap will be fulfilled in this study. Secondly, the role of paradoxical leadership in promoting the adaptive performance of new-generation employees will be thoroughly demonstrated, and strengthened so as to systematically understand paradoxical leadership. In this study the role and effect of paradoxical leadership in the adaptive performances of new-generation employees are well analyzed, and the applicable objects and positive aftereffects of the local paradoxical leadership, together with the precursor content to discuss the adaptive performance based on the leadership level, are further extended. Thirdly, this study is focused on the self-determination theory to explore the role of paradoxical leadership in adaptive performances of new-generation employees, and we analyzed the effective principle of paradoxical leadership work with the idea of the generation and transformation of harmonious work passion, which is helpful to understand the essence and effective mechanism of paradoxical leadership. Fourthly, the core self-evaluation of employees into the research framework is included, and the boundary conditions that paradoxical leaders' roles play are clarified, which is very conducive to promoting the situational research of paradoxical leadership and providing the theoretical inspiration and practical guidance for the enterprises to improve adaptive performances of new-generation employees.

Overall, firstly, based on the theoretical research of paradoxical leadership, adaptive performance, harmonious work passion, core self-evaluation and new-generation employees, this paper puts forward research assumptions. Then, the research design and analysis of the results are discussed, followed by the discussion section, in which the theoretical and practical implications of this paper are elaborated, and the current limitations and future prospects are pointed out. Finally, the conclusions of this paper are drawn. Therefore, this paper hopes to sound the alarm in advance for enterprises in the post-epidemic era to guard against the emergence of toxic working environments. This enriches the theoretical achievements of leaders guiding employees to take the road of healthy development and create good performance. Similarly, it also helps enterprises adapt to the post-epidemic era, meet the dual development requirements of organizations and employees and achieve sustainable development.

## 2. Literature Review and Research Hypotheses

### 2.1. Paradoxical Leadership and Adaptive Performance

Facing the increasingly changing complex external organizational environment in the post-epidemic era, management leaders have to deal with quite a lot of contradictions and uncertainties [33]. How to effectively respond to the dilemma and achieve a win-win situation poses new and severe challenges to leaders. At the same time, in view of new-generation employees within organization increasingly becoming the main force [8], leaders inevitably have to solve the conflicts that arise between the diverse needs of new-generation employees and the organizational needs [34]. Paradoxical leadership, as a seemingly contradictory but actually coordinated and unified leadership type, can deal with the aforementioned problem through a dialectical and unified perspective to solve the complex situation and contradictions in the organization, which can surely bring new ideas to the efficient management activities of leaders [22,35]. The current scholars, on the basis of the paradox research combined with the Chinese philosophy theory of Yin and Yang, have formally put forward the concept of paradoxical leadership. There are five significant characteristics: (1) as far as the center is concerned, the combination of self and others should be highlighted; (2) as far as the relation is concerned, the loose and close relations should be co-existent; (3) as far as the strategy is concerned, the commonness as well as the particular characteristics should both be well considered; (4) as far as the rule is concerned, the strictness and the flexibility are equally important; (5) as far as the process is concerned, the control and the encouragement should be of parallel significance [15]. The paradoxical leadership concept formation and its scale development could definitely provide the premise for the in-depth exploration of the paradoxical leadership, but the

internal function process and application scope of paradoxical leadership still need to be further explored so as to enrich the research of the paradoxical leadership theory.

The employees, who are now living in the dynamic changing environment while working in the organization, either internally or externally, in the post-epidemic era, also have to meet with the problems: either by changing themselves or adapting. For example, in view of the fact that the growing use of instant messaging is associated with perceived technological complexity, overload and intrusion of employees, and the effects on their job performance and happiness [36], how should employees properly cope with the technical pressure brought by the widespread use of instant messaging? The adaptive performance, which is proposed in this essay, reflects the enterprise's urgent needs for adaptive employees in the current complex changeable environment. Conversely, it increases a lot of new management requirements for the traditional performance content [7]. The adaptive performance, as a concept which is presented on the basis of the previous task performance and peripheral performance [37], means the individual adaptive behavior when facing the complex changeable organizational environment [5]. The concept of adaptive performance reflects the employees' behavior as having been adjusted in order to cope with dynamic changes. In other words, the employees' performances will be inspected and examined in the consideration of the employees' behavior. The new-generation employees are those who were born after 1980 and 1990 [9,20]. This labor force group, as a key labor force in the reform and development of enterprises, has the potential to meet any challenges but also have disadvantages in encountering any setbacks. Thus, it is a very important research topic for human resource management, which focuses on how to enable this labor force group to have a full role in the enterprises so as to stipulate their adaptive behavior to be maximumly advantageous.

As a newly-emerging leadership style with location-specific characteristics, the in-organization paradoxical leadership mentioned in the essay encourages leaders to reshape their work to improve their task performance [38], and is good at using the Doctrine of the Mean to deal with the complex and changeable problems in the organization from the perspective of contradictory integration [15,16]. It sets a benchmark for employees who are unhappily stuck in the unpredictable environment, and helps to flexibly deal with these problems. The new-generation employees are characterized by strong learning and imitation capabilities [8]. They can quickly learn, imitate and improve their own critical thinking abilities and perception levels when they are faced with the positive flexible behavior mode, and are capable to improve their overall adaptive performance to a certain degree when dealing with the dynamic environment and task changes. The dialectical-unified problem-solving mode of paradoxical leadership can simultaneously shape the organizational circumstances, or so-called situation, featured with open-mindedness and boundary restrictions, and construct a harmonious and balanced organizational atmosphere in which the employees' working-proficiency, working-initiative and working adaptability can all be upgraded [15]. Therefore, the employees' adaptive performance will be also effectively accelerated.

In addition, as far as the five specific characteristics of paradoxical leadership are concerned, it is found that those five characteristics are highly and compatibly conformed to the needs of new-generation employees, which can play a very important role in the improvement of the adaptive performance. Specifically speaking, as far as the center is concerned, the close combination between the employee's own self and his/her counterparts can well maintain the leadership authority as well as the employees' own advantages [15]. Although the new-generation employees have certain exploration and adaptability, they tend to have more uncertainties in terms of their job engagement and adaptive behavior in working practice [18]. However, paradoxical leadership can well exploit the employees' advantageous potential, arouse their great respects, foster their commitment to execution [17], enhance their stability of job engagement and adaptive behavior, and strengthen their adaptive performance further. Secondly, as for the relations, the close relation and the loose relation co-exist well, only to maintain a just-right distance between the leadership

and the employees [15]. It is known that the leaders and employees are structured at the different organizational levels. Supposing that the relations between the leadership and the employees is too close, it will generate a sense of over-confidence, feeling in favor and tendencies to be overly prideful [39,40]. If the relation is kept too far apart, the employees' working sensitivity tends to be reduced [41], and consequently the employees working enthusiasm will be damaged. The paradoxical leadership featured with the close–loose interpersonal relations can offer new-generation employees proper space and emotional warmth, which are especially needed in their development, and enable them to maintain sensitivity and enthusiasm in their work, which can purposefully improve their adaptive performance. Thirdly, in terms of the strategy, both the commonness and the particular characteristics should receive equal attention [15]. The principle called Competency-based Instructing System must be carried out while the employees' aptitudes are considered in equal treatment. This can create an atmosphere in which there are fulfilled with favorability and impartiality when paradoxical leadership is applied, and the development opportunities required by the uniqueness of employees is sufficiently considered. Thus, it will be very helpful for the employees to build mutual trust among the team members and formulate a healthy reliable working environment [42] in order to enable new-generation employees to be better integrated into the organization. Additionally, new-generation employees, depending upon the strength of the team members and team resources, will more effectively meet the challenges of their working task changes, thus improving adaptive performance. Fourthly, the strictness and the flexibility should be presented together in terms of work requirements [15]. The paradoxical leadership always cherishes the high expectations and requirements for employees' ability to overcome the difficulties, such as the problem-solving ability for any changes they meet. Meanwhile, paradoxical leadership would like to offer enough support and encouragement for the employees [43], which can relatively improve the employees' workplace satisfaction and psychological security. It is thought to be very important for the organization to stimulate the employees' positive emotion, as well as the initiative of new-generation employees' adaption of themselves to the internal and external changes in the organization [44–46]. Moreover, the already-conducted studies have testified that the employees, who are working and living in an atmosphere where the leaders and the employees are getting along well and the latter can receive timely support from the former, would like to have more capabilities and show willingness to perform their adaptive behaviors [47]. Fifth, the control and the encouragement should be autonomously balanced whenever the process is involved [15]. In an organization, only one pole of contradictory leadership, such as excessive regulatory control or overly flexible authorization, is not conducive to the smooth operation of organization and the active follow-up of employees, but the coexistence of seemingly opposite types of leadership can achieve long-term prosperity of organization and good response of employees [48]. Paradoxical leadership sets clearly up overall goals so as for the employees to reduce their sense of confusion at work. Paradoxical leadership also gives employees enough autonomy for their self-exploration and self-leadership, achieving the flexible allocation between the authority and the responsibilities, which can help new-generation employees focus on their goals whenever any emergencies occur, and properly deal with these emergencies in time [49]. Therefore, paradoxical leadership can not only stimulate the potential of new-generation employees at work but also help to improve their adaptability. This could not only help their shortcomings during work, but could also facilitate the implementation of their adaptive behaviors. Accordingly, the hypothesis is presented below.

**H1.** *Paradoxical leadership has a significant positive impact on adaptive performances of new-generation of employees.*

### 2.2. Mediating Role of Harmonious Work Passion

The work passion is generated from the general passion. Along with the application of general passion in organizational situations, there brings about the in-depth insights into the research on the employees' behavior and performance. It is found that controlled moti-

vation is related to compulsive work, which gives rise to job burnout, while autonomous motivation is related to work enthusiasm, which fails to produce unpleasant feelings [50]. The Binary Model of Passion—that is, the harmony and the forced passion—was first set up by Vallerand et al. [51] on the basis of whether self-autonomous standards can be internalized. The harmonious work passion is a kind of positive energy produced at work, which is present in their positive psychological state, which will be embodied in their love, their autonomous thinking and their time/energy contribution of their voluntary willingness when they are at work, which includes the individual emotion, cognition and motivation [25,26,52].

It is pointed out in the self-determination theory that when sufficient and reliable support information is provided from outside, it can satisfy the psychological needs for employees' relationship, competence and autonomy so as to stimulate the upsurge of their passion for harmonious work [51,53]. In the working circumstances, the leadership behavior, as one of the external factors that are mainly observed and concerned by the employees, will always generate great impact upon the employees. The appropriate behavior of the leadership is of great importance to stimulate the employees' harmonious work passion [54]. The paradoxical leadership conducts management activities in accordance with the philosophy of Yin and Yang [15]: all these elements, such as emotional warmth (temperature), cognitive rationality (attitude) and behavior coordination (depth), which are seemingly contradictory but factually unified human behavior, can better serve the high-level psychological needs of new-generation employees.

Specifically speaking, the first is that paradoxical leadership can not only keep appropriate distance with employees and maintain the proper close contact, but also treat employees equally and respect personality development [15], which gives new-generation employees a sufficient sense of the interactive fairness and harmonious relationships, to a certain extent [55]. The maintenance of the appropriate distance and equal treatment can create a fair and healthy organizational environment, which can avoid the malicious competition and conflicts, easily build a good colleague relationship and enhance the team trust and cohesion among new-generation employees [42]. The close contact and respect for personality development can facilitate and strengthen the communication among new-generation employees, who are capable of helping leaders duly understand the employees' needs and motivation, with the easiness to build up a good relationship between leaders and subordinates, thus enhancing the emotional cohesive power and follow-up commitment of new-generation employees [56]. The warmth of the humanistic care of paradoxical leadership can meet the needs of new-generation employees and maintain a harmonious relationship between the leaders and colleagues within the organization. The next is that paradoxical leaders advocate to combine the personal self with other-people orientation, so as to simultaneously maintain the requirement standards and the flexible problem-solving treatment [15]. Therefore, the sense of successful working achievement and individual value realization, in the minds of new-generation employees, will be relatively enhanced. The personal self-orientation and the requirement standards can apparently constrain new-generation employees' behavior [57], prevent them from deviating away from the organizational common goals and help them to accumulate their working experiences. The notion of other-people orientation and the permission to flexibly deal with the problems can surely affirm the value realization and personal contribution of new-generation employees, cultivate their willpower to exploit their potentialities and perseverance to tap their own career development [58], and finally increase recognition and expectation of their own abilities. The attitude of the paradoxical leadership can meet the competency needs of new-generation employees to complete their work with high quality and efficiency. Finally, paradoxical leadership pays attention to both control and autonomy [15], which, to some extent, will augment the sense of responsibility and autonomy of new-generation employees. New-generation employees will be equipped with back-up supporting resources via the decision-making control. Therefore, decision-making risks by new-generation employees can be decreased, their sense of depression or frustration can also be alleviated, and their

sense of mastery and initiatives in work will be consequently enhanced. To empower the autonomy to new-generation employees to offer them a chance to have a free working-play and self-management stimulates their enthusiasm to participate in the decision-making process and their willingness to explore any new changes, which improves their creativity and initiative, and encourages them to maximize their own effectiveness [59]. As for the depth of paradoxical leadership, it will satisfy the independent needs of new-generation employees to display their talents in a supportive atmosphere.

Harmonious work passion means a more accommodative mental process that guides people to love their job, acquire key resources and get into the swing of their work [60]. The harmonious work passion inspired by the paradoxical leadership may also affect the adaptive performance of new-generation employees. First of all, in terms of the positive emotional attributes, the new-generation employees with harmonious work passion often have a high sense of organizational belonging and love [61], which will strengthen their community consciousness with a shared future with the organization, enhance their sense of responsibility and work vitality in the situations which are always changing, enable them to willingly support the organizational management changes and innovative development, and prompt them to actively participate in and focus on their work tasks during any dynamic changes. Secondly, in terms of the positive cognitive attributes, new-generation employees, if they have harmonious work passion, are always featured with the high sense of work significance and worth [51]. With that being the case, the scope of personal cognition can be persistently augmented [62] and the flexibility to mobilize all resources will be enhanced to control the current actions [28], which will lay a solid foundation to aptly predict and respond to any changes that occur. Finally, as far as the autonomous motivative attributes are concerned, new-generation employees who are equipped with harmonious working passion tend to get their confidence and patience strengthened when facing any uncertainties in the situation changes whether they happen in or outside the organization, thus they are actively looking for diversified solutions to deal with difficult problems [63,64] so as to provide more solutions to adapt to the internal and external changes of the organization.

Further, harmonious work passion can facilitate the efficient connections between paradoxical leadership and adaptive performances of new-generation employees. The relevant studies have already proved that leadership behavior not only directly affects the employees' behavior, but also indirectly affects their behavior and performance via influencing their cognitive and emotional status [65]. Paradoxical leadership can benefit new-generation employees to obtain an independent platform, perceive the support and respect of their superiors and colleagues, control the work and accumulate successful experience. This can, finally, promote new-generation employees to internalize their work independently and stimulate them to have higher passion for their harmonious work. The harmonious work passion can strengthen the drive of new-generation employees' good adaptive performance, so that they are actively committed to adaptive performance improvement activities. Therefore, it is speculated that the harmonious work passion is an important intermediary mechanism in the relationship between paradoxical leadership and adaptive performance in new-generation employees. Moreover, it is assumed in this paper that:

**H2.** *Harmonious work passion plays a mediating role between paradoxical leadership and adaptive performance.*

### 2.3. Moderating Role of Core Self-Evaluation

Core self-evaluation integrates the following four elements: Self-efficacy, Emotional stability, Control point and Self-esteem, which are regarded as the individual's overall judgment about their own strength and life meaning [31]. As a higher-order individual trait, the core self-evaluation is the basis of individual situational assessment and produces significant effects on individual emotional cognitive state and work behavior performance. Core self-evaluation can play an important role in regulating the relationships in many

different variables [66]. The influences, due to the situational factors on the individual motivation and behavior, can be influenced by the individual trait differences. The individual traits/idiosyncrasy either promote or inhibit the individual positive emotions, attitudes or behaviors. However, the crucial point is whether the individual can correctly identify the situation connotation and then adjust themselves to be adapted with the situation [27,29]. Research shows that core self-evaluation is of great significance in promoting employees' positive work experience, motivation internalization and active behavior [67]. Therefore, it is predicted, in this essay, that the new-generation employees with higher core self-evaluation consciousness can actively interpret the dual characteristics of the paradoxical leadership situation, which is a key individual factor to stimulate the role of harmonious work passion with the idealistic exploitation of paradoxical leadership.

Specifically, new-generation employees with high core self-evaluation tend to have the high stress tolerance and the sense of self-regulation [68]. They will maintain their objective cognitive evaluation and positive coping strategies; when they face the boundary control of paradoxical leaders, they will always internalize the work requirements into their work motivation so as to effectively play the positive role of paradoxical leaders on their harmonious work passion. Conversely, new-generation employees with high sense of the core self-evaluation are more sensitive to positive scenarios [69]. In the face of the open supports provided by their leaders, they can detect and capture the favorable resources in a timely fashion and can then flexibly apply them into their work while transforming the positive support into advantages of their own [70]. Therefore, their own excellent abilities and organizational commitment can be well strengthened to achieve the result that their harmonious work passion is idealistically stimulated. It can be speculated, thereout, that paradoxical leadership can not only make up for their lack of frustration and temporary drive, but can also strengthen their advantages of sensitivity and professionalism. When paradoxical leadership acts on the new-generation employees with high core self-evaluation, it makes it more likely to produce harmonious work passion. On the contrary, as far as the new-generation employees with low core self-evaluation within the organization are concerned, the boundary control of the leaders will bring them limited work interpretation and their internalization of external motivation will be consequently affected, which is not conducive to stimulating their harmonious work passion. Similarly, new-generation employees with low core self-evaluation do not easily detect the work support given by their superiors, which is not conducive to the construction of their own harmonious work passion. Therefore, it is not easy for paradoxical leadership to have a significant impact on the new-generation employees with low core self-evaluation. In conclusion, this paper makes the following assumptions:

**H3.** *Core self-evaluation positively moderates the relationship between paradoxical leadership and harmonious work passion of new-generation employees.*

Combined with H2 and H3, paradoxical leadership can affect adapt the performance of new-generation employees via the harmonious work passion, while the stimulation of paradoxical leadership to the harmonious work passion will be positively affected by the core self-evaluation of new-generation employees. Therefore, it can be further predicted that the mediation effect of harmonious work passion between paradoxical leadership and adaptive performance of new-generation employees can surely be regulated by the core self-evaluation. Specifically, under the condition that new-generation employees are supposed to have higher core self-evaluation, the relationship between paradoxical leadership and harmonious work passion will be stronger. The promotion of new-generation employees' harmonious work passion makes it easier to stimulate their motivation to deal with organizational changes, and to optimize their adaptive performance. However, on the contrary, new-generation employees who feature weak senses of the core self-evaluation will always ignore the positive role of the paradoxical leadership, and can even weaken the role of paradoxical leadership in stimulating harmonious work passion. Employees with weaker senses of core self-evaluation will finally be manifested with lower-level adaptive

performance due to the lack of their positive confident attitudes and unwillingness to face the organization boundary control and open authorization. In accordance with the analysis above, the following assumptions are to be presented in this paper:

**H4.** *The mediating effect of harmonious work passion between paradoxical leadership and adaptive performance of new-generation employees is moderated by core self-evaluation. The higher the core self-evaluation, the stronger the mediating effect; otherwise, the weaker is true.*

According to the above, the theoretical model of this study is illustrated in Figure 1.

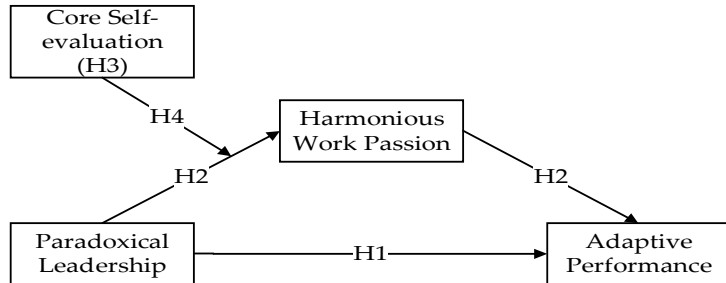

**Figure 1.** Theoretical model (Authors' proposal).

### 3. Method

#### 3.1. Study Sample

View-points such as Podsakoff's have been adopted in this study [71]. For example, the questionnaire survey is conducted among the new-generation employees in the technology enterprises in Beijing and Tianjin at different time points so as to avoid any deviation problems due to the same or common method. Before the survey, the head of the human resource department where the survey is to be conducted is well communicated to in order to confirm the would-be-surveyed subjects, and the coding procedures are strictly programmed to purposefully make sure that the matching of information can be obtained at different time points. In order to improve the enthusiasm and willingness of subjects to participate in the survey, the participants can get small rewards after reviewing the answers. In addition, the participants are well informed by the on-site research team members in advance that the confidential questionnaire is only used for academic research, together with comprehensive explanations. At the first time-point, the questionnaires with statistical variables, paradoxical leadership, harmonious work passion and core self-evaluation were distributed to 629 subjects. As a consequence, 583 questionnaires were ensured to be valid. An interval of 3 months later, the questionnaires of adaptive performance were collected from the subjects who had participated in the survey for the first time. After eliminating invalid answers and lost samples, 519 valid matched questionnaires were finally obtained. The demographic characteristics of the sample are shown in Table 1.

**Table 1.** Sample Demographics.

| Statistical Variables | Category | Frequency | Percentage |
|---|---|---|---|
| Gender | Male | 247 | 47.59% |
| | Female | 272 | 52.41% |
| Age | Less than 26 | 156 | 30.06% |
| | 26–30 years old | 160 | 30.83% |
| | 31–35 years old | 122 | 23.51% |
| | 35–40 years old | 81 | 15.60% |
| Education | Junior college | 59 | 11.37% |
| | Undergraduate | 188 | 36.22% |
| | Master | 197 | 37.96% |
| | Doctor | 75 | 14.45% |

*3.2. Measurement Tools*

The mature scales from previously studied literature are used in this essay. Furthermore, the procedures for the translation and the back-translation of those mature scales are seriously conducted, only purposefully to translate the English scales into Chinese. With the exception of the control variables, the variables studied were scored by the Likert5 Grade Method, in which 1 stands for "very inconsistent" and 5 stands for "very consistent".

The measurement for paradoxical leadership is referred to the 22-question scale developed by the Zhang et al. [15]. The method for the measurement is called the indirect reports from the subordinates, with such example topics as "My direct supervisor can decide major issues and give employees the right to decide secondary issues" and "My direct supervisor communicates with employees equally and adjusts the communication methods according to different characteristics", and the Cronbach's $\alpha$ is 0.955. The measurement of harmonious work passion refers to the seven questions by Vallerand [51], such as "Work gives me a colorful experience", "New discoveries in the process of work make me more interested in my work", and the Cronbach's $\alpha$ is 0.879. The measurement of adaptive performance draws lessons from Pan et al. [72] and uses the 9-question scale compiled by the Griffin [3], with example questions such as "I can effectively respond to changes in how core tasks are completed" and "I can flexibly handle overall changes in the organization", with 0.889 as Cronbach's $\alpha$. Core self-evaluation is measured with reference to the 12-question scale developed by Judge et al. [31], in which there are 6 forward and 6 reverse questions, for instance: "I am able to successfully complete tasks" and "Sometimes I will feel frustrated", with 0.935 as its Cronbach's $\alpha$. Particular measurement items are provided in Appendix A, Table A1. On the basis of the previous studies in terms of adaptive performance [73], age, gender and education background are selected as control variables to reduce the effect on the research results.

## 4. Results

*4.1. Common Method Bias*

The SPSS 26.0 and the AMOS 24.0 are adopted to process the data in this research. In consideration that the data used in this study were acquired from the employee self-reported information, even if the information survey is conducted at different time-points, and together with methods such as confidential answers and the addition of reverse questions, it is still thought very much necessary to implement further tests and investigations in order to suppress the common method biases. Therefore, the Harman Single-factor Test was used in this study [71], and we consequently found that the explanatory degree of the first component obtained without rotation is 34.69%, which is lower than 40% as the critical value. Thus, it is indicated that the common method bias problem is within the controllable range.

*4.2. Descriptive Statistical Analysis*

The mean, standard deviation and correlations of the main variables involved in this study are listed in Table 2. Paradoxical leadership is positively correlated with adaptive performance (r = 0.449, $p < 0.01$) and harmonious work passion (r = 0.447, $p < 0.01$). Harmonious work passion is positively correlated with adaptive performance (r = 0.410, $p < 0.01$). These results provide the preliminary supports for the next-step hypothesis test.

**Table 2.** Mean, standard deviation and correlation coefficient.

| Variables | Mean | SD | 1 | 2 | 3 | 4 | 5 | 6 |
|---|---|---|---|---|---|---|---|---|
| Gender | 0.524 | 0.500 | | | | | | |
| Age | 2.247 | 1.049 | 0.022 | | | | | |
| Education | 2.555 | 0.875 | −0.039 | 0.067 | | | | |
| Paradoxical leadership | 3.260 | 0.777 | 0.046 | −0.017 | 0.096 * | | | |
| Harmonious work passion | 3.353 | 0.796 | −0.026 | −0.037 | 0.061 | 0.447 ** | | |
| Core self-evaluation | 3.237 | 0.817 | −0.001 | −0.009 | 0.096 * | 0.565 ** | 0.447 ** | |
| Adaptive performance | 3.234 | 0.883 | −0.052 | −0.020 | 0.066 | 0.449 ** | 0.410 ** | 0.371 ** |

Note: N = 519, ** $p < 0.01$, * $p < 0.05$.

### 4.3. Confirmatory Factor Analysis

The composite reliabilities of paradoxical leadership, harmonious work passion, core self-evaluation and adaptive performance were 0.920, 0.879, 0.935 and 0.823, respectively, all of which were greater than 0.7. Their average variance extracted values were 0.696, 0.511, 0.545 and 0.608, respectively, all of which were greater than 0.5, and all are exceedingly more than the standard of convergence validity. In addition, confirmatory factor analysis (Table 3) shows that among the factor models, the four-factor model has the best fitting effect in each factor model: $\chi^2/\mathrm{df}$ is between 1 and 3, RMSEA is lower than 0.05, AGFI is higher than 0.8, GFI, and NFI and CFI are higher than 0.9, which shows that each variable has good discrimination validity and is suitable for further analysis.

**Table 3.** Confirmatory factor analysis.

| Model | $\chi^2/\mathrm{df}$ | RMSEA | GFI | AGFI | NFI | CFI |
|---|---|---|---|---|---|---|
| Four-factor model | 1.239 | 0.022 | 0.903 | 0.894 | 0.916 | 0.983 |
| Three-factor model | 2.008 | 0.044 | 0.797 | 0.778 | 0.864 | 0.927 |
| Two-factor model | 2.824 | 0.059 | 0.721 | 0.695 | 0.809 | 0.867 |
| Single-factor model | 3.052 | 0.063 | 0.703 | 0.676 | 0.793 | 0.850 |

Note: The four factor model is paradoxical leadership, harmonious work passion, core self-evaluation, adaptive performance. The three factor model is paradoxical leadership + harmonious work passion, core self-evaluation and adaptive performance. The two factor model is paradoxical leadership + harmonious work passion + core self-evaluation and adaptive performance. The single factor model is paradoxical leadership + harmonious work passion + core self-evaluation + adaptive performance.

### 4.4. Empirical Results

In the main effect test, the hierarchical regression method was used to analyze the relationship between paradoxical leadership and adaptive performance of new-generation employees. To be specific, the adaptive performance factor was used as the dependent variable while gender, age and education as control variables were included into the regression model. As for the regression results, please see Table 4 Model 1. Further on, the paradoxical leadership factor was added as the independent variable; as for the final results, please see Table 4 Model 2. Paradoxical leadership is significantly featured with the positive influence on the new-generation employees' adaptive performance ($\beta = 0.450$, $p < 0.001$) in the assumption that H1 is reasonably verified.

The mediating effect was tested by the means of Baron and Kenny [74] Step-by-step Method. First of all, the main effect test indicated that H1 achieved validation. Secondly, the relationship between paradoxical leadership and harmonious work passion was tested. The results were shown in Table 4 Model 6. It was proven that paradoxical leadership had a significantly positive effect on harmonious work passion ($\beta = 0.447$, $p < 0.001$). Then, the relationship between harmonious work passion and adaptive performance was tested, with the results clearly shown in Table 4 Model 4, where harmonious work passion significantly positively affected adaptive performance ($\beta = 0.407$, $p < 0.001$). Finally, the mediating variable was added based on Model 2, and the results were detailed in Table 4 Model 3. The predictive effect of paradoxical leadership decreased ($\beta = 0.335$, $p < 0.001$), and the predictive effect of the harmonious work passion was significant ($\beta = 0.258$, $p < 0.001$). Therefore, harmonious work passion has a partial mediating role between paradoxical leadership and adaptive performance of new-generation employees, thus assuming that H2 is supported.

In the moderating effect test, the interaction item between paradoxical leadership and core self-evaluation was multiplied after centralization. When the harmonious work passion factor was used as a dependent variable, firstly, the control variables were included in the regression model, and the results were detailed in Table 4 Model 5. Secondly, the paradoxical leadership variable was added, and the results were detailed in Table 4 Model 6. Then, the core self-evaluation variable was added, and the results were detailed in Table 4 Model 7. Finally, the interaction item of paradoxical leadership and core self-evaluation were added, and the results were shown in Table 4 Model 8, which indicates that the prediction effect of

the interaction item is shown to be significant ($\beta = 0.231$, $p < 0.001$). In order to clearly present the moderating effect of core self-evaluation, this study added and subtracted a standard deviation based on the mean value of core self-evaluation to distinguish different levels and drew a moderating effect diagram (See Figure 2 for details). Figure 2 shows that the slope of the high core self-evaluation was greater than that of the low core self-evaluation, where core self-evaluation has positive moderating effect, assuming H3 holds. The moderated mediating effect was tested by using the PROCESS and the results were detailed in Table 4. When core self-evaluation was low, the indirect effect of paradoxical leadership on adaptive performance through harmonious work passion was 0.047, and the 95% confidence interval was $[-0.001, 0.106]$. When core self-evaluation was high, the indirect effect of paradoxical leadership on adaptive performance through harmonious work passion was 0.151, and the 95% confidence interval was $[0.076, 0.244]$. In reference to Hayes [75], through the auxiliary judgment of INDEX, INDEX was 0.065 and its confidence interval was $[0.024, 0.114]$, excluding 0, which can show that core self-evaluation moderates the mediating effect of harmonious work passion between paradoxical leadership and adaptive performance of new-generation employees, and that H4 holds.

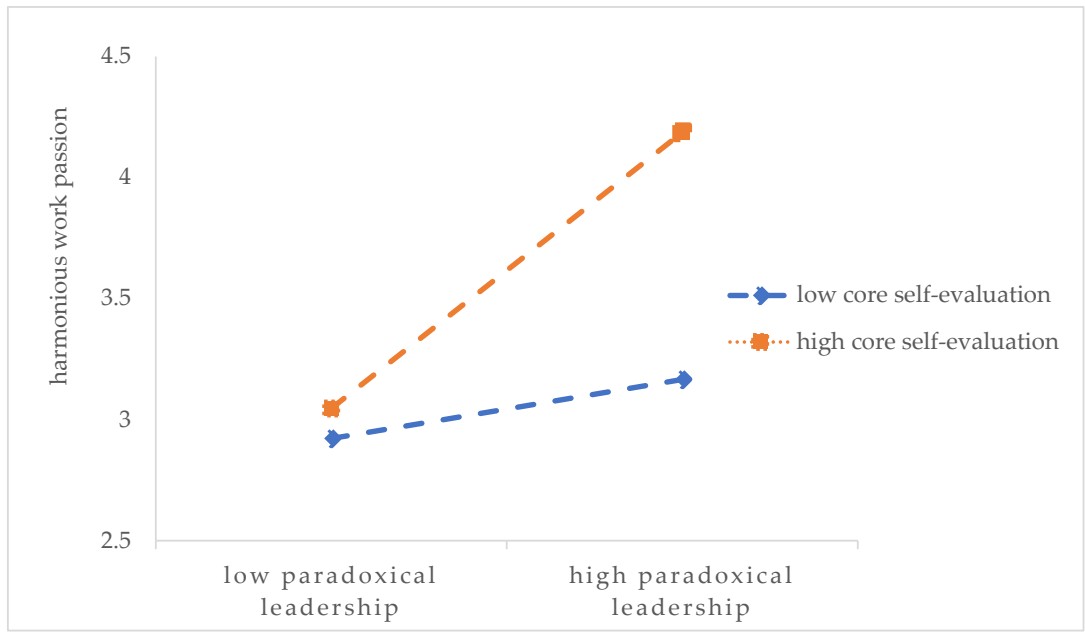

**Figure 2.** The moderating effect of core self-evaluation (Authors' proposal).

**Table 4.** Results of hierarchical regression analysis.

| Variables | Adaptive Performance | | | | Harmonious Work Passion | | | |
|---|---|---|---|---|---|---|---|---|
| | **Model 1** | **Model 2** | **Model 3** | **Model 4** | **Model 5** | **Model 6** | **Model 7** | **Model 8** |
| Gender | −0.049 | −0.072 | −0.060 | −0.039 | −0.023 | −0.046 | −0.038 | −0.018 |
| Age | −0.024 | −0.013 | −0.005 | −0.007 | −0.040 | −0.030 | −0.029 | −0.036 |
| Education | 0.066 | 0.021 | 0.016 | 0.040 | 0.063 | 0.019 | 0.007 | 0.008 |
| Paradoxical leadership | | 0.450 *** | 0.335 *** | | | 0.447 *** | 0.288 *** | 0.338 *** |
| Harmonious work passion | | | 0.258 *** | 0.407 *** | | | | |
| Core self-evaluation | | | | | | | 0.283 *** | 0.293 *** |
| Interactive item | | | | | | | | 0.231 *** |
| $R^2$ | 0.007 | 0.207 | 0.260 | 0.172 | 0.006 | 0.203 | 0.258 | 0.307 |
| $\Delta R^2$ | 0.007 | 0.200 | 0.053 | 0.165 | 0.006 | 0.197 | 0.055 | 0.049 |
| F | 1.275 | 33.617 *** | 36.114 *** | 26.629 *** | 1.027 | 32.779 *** | 35.631 *** | 37.856 *** |

**Table 4.** *Cont.*

| Variables | | Adaptive Performance | | | | Harmonious Work Passion | | |
|---|---|---|---|---|---|---|---|---|
| | | Model 1 | Model 2 | Model 3 | Model 4 | Model 5 | Model 6 | Model 7 | Model 8 |
| the moderated mediating effect | moderator variable | level | effect | SE | Boot 95% CI | INDEX | SE | Boot 95% CI |
| | core self-evaluation | low | 0.047 | 0.028 | [−0.001, 0.106] | 0.064 | 0.023 | [0.024, 0.114] |
| | | high | 0.151 | 0.043 | [0.076, 0.244] | | | |

Note: *** $p < 0.001$; the interaction term = Paradoxical leadership × Core self-evaluation; the CI is the confidence interval; the high/low of the moderator variable means one standard deviation above/below the mean.

## 5. Discussion

This study builds a moderated mediation model based on the self-determination theory in order to clarify why, how and when paradoxical leadership can promote the adaptive performance of new-generation employees in the post-pandemic era. First of all, this study demonstrates that paradoxical leadership is a momentous situational component to improve adaptive performance of new-generation employees, which responds to prior propositions that shed light on the research of new-generation employees [9] and explores adaptability improvement strategies from the sentiment of paradox [76]. In previous studies, it has been discovered that individual and situational characteristics can predict individual adaptive performance [7], whereas the relationship between paradoxical leadership and adaptive performance of new-generation employees still needs further testing. The paradox viewpoint of corporate sustainability [77] points out that the paradox optic angle enables managers to realize rival sustainable development goals in synchrony, and provides room for contributing to sustainability. Paradoxical leadership, as an important situational element, is deft at using contradiction integration strategies based on paradox cognition. Paradoxical leadership can not only attract the new-generation employees with stronger subjective initiative to learn and then enhance their adaptability, but also be in line with the concerns of new-generation employees with stronger self-awareness and strengthen the implementation of adaptive behavior, thus achieving a high level of adaptive performance. Empirical results show that paradoxical leadership has a positive effect in predicting the adaptive performance of new-generation employees. These research results correspond with the conclusion of Zhang et al. [15], which affirms the positive influence of paradoxical leadership on employees. In the meantime, the paradox standpoint is adopted to deal with various conflicts of sustainability activities in the post-epidemic era, which provides a new perspective to work out sustainability problems. In addition, Fang et al. [8] analyzed the effect of leadership on the passive behavior of new-generation employees. On the contrary, this study explores the predictive factor of stimulating adaptive performance of new-generation employees from the perspective of paradoxical leadership. This study focuses on the micro-level adaptive performance of new-generation employees and enriches the research level between leadership and performance; conversely, it is conductive to cultivate organizational resilience and enhance sustainable performance.

Secondly, this study validates the mediating role of harmonious work passion between paradoxical leadership and adaptive performance of new-generation employees, and responds to the appeal for research on how the paradoxical perspective of leadership produces outcomes in organizations [78]. Paradoxical leadership has an impact on new-generation employees and this study speculates that paradoxical leadership may trigger the related psychological processes of new-generation employees, which affects their adaptive performance. Based on self-determination theory, this study examines the role of harmonious work passion between paradoxical leadership and adaptive performance of new-generation employees. The empirical results confirm that harmonious work passion plays a partial mediating role between paradoxical leadership and adaptive performance of new-generation employees, which manifests that paradoxical leadership can not only directly predict adaptive performance of new-generation employees, but also indirectly

predict adaptive performance of new-generation employees through harmonious work passion. This result is similar to the framework of "leadership-work passion-employee work results" proposed by Fang et al. [8], who state that leadership can affect employees' work passion level and subsequently act on employees' work results. What is more, the results of this study demonstrate the transmission process of paradox thinking, strategy and behavior on the leadership level in organizations, which enriches the strategic basis for resolving the complexity of sustainable problems.

Thirdly, the influence of leadership on employees is not exactly the same, and it may vary according to different characteristics of employees. Therefore, this study also discusses the role boundary of paradoxical leadership in combination with core self-evaluation. Previous studies on the role boundary of paradoxical leadership focused on team factors, and less on personal traits. Therefore, this study explores the role of core self-evaluation in the relationship among paradoxical leadership, harmonious work passion and adaptive performance of new-generation employees. The results of data analysis confirm the moderating effect of core self-evaluation. In other words, core self-evaluation not only plays a moderating effect between paradoxical leadership and harmonious work passion, but also adjusts the mediating effect of harmonious work passion. This result is in accordance with previous studies. That is to say, when employees have positive personality traits, the positive effect of leaders on employees' work response can be reinforced [32]. This study shows the reaction to the viewpoint of Li et al. [22], which is that in the process of employees' commitment to workplace activities, besides leaders playing a key role, we should also pay attention to the role played by personal characteristics in this process. Moreover, it further optimizes the actual study framework to better put into practice sustainable management in the post-epidemic era.

## 6. Research Implications and Prospects

### 6.1. Theoretical Implication

The theoretical implications of this study mainly have the following four aspects. First, with new-generation employees as the research object, the improvement mechanism of adaptive performance in the post-pandemic era is well discussed. The uniqueness and the necessity of the adaptive performance for the sustainable development of enterprises are highlighted, and the relevant researches are also conducted for expansion regarding performance management.

Secondly, the positive impact of the paradoxical leadership on the employees' adaptive performance is validated in this study. The research conducted by the previous scholars was mainly focused on authentic leadership [10], servant leadership [10,11], self-leadership [12,13] and shared leadership [14], which are all featured within the western cultural background of the single leadership behavior on the adaptive performance. However, in this study introduced paradoxical leadership, which is characterized the Chinese cultural situation and the epochal development trend. The influence of dual-characteristic leadership on new-generation employees' adaptive performance are analytically discussed so as to effectively enrich the researches of the adaptive performance and to also effectively respond to the appeals of the previous scholars calling to explore the mechanism of paradoxical leadership and employees' adaptive behavior.

Thirdly, in accordance with the self-determination theory, harmonious work passion integrated with cognition, emotion and motivation is herein introduced as the mediating variable, so as to reveal the "black box" of the relationship between the paradoxical leadership and new-generation employees' adaptive performance. In comparison to previous research on the mediating path of paradoxical leadership to performance from a single cognitive perspective, the mediating path mechanism of the paradoxical leadership on adaptive performance is apparently and greatly enriched, which provides a comprehensive explanation and strong support for paradoxical leadership to stimulate new-generation employees' harmonious work passion, thereby producing adaptive performance.

Finally, the moderating effect of core self-evaluation, such as a high-order personality trait, is further analyzed, which provides a basis for the effective clarification of the boundary conditions of the paradoxical leadership's impact on new-generation employees' adaptive performance. The positive effect of core self-evaluation in organizational management is well verified in this study while the boundary conditions of the paradoxical leadership influence mechanism are well enriched. Furthermore, the consideration of paradoxical leadership and the core self-evaluation of new-generation employees at the same time is proven to be more helpful to clarify the influence of leadership on different employee groups.

### 6.2. Practical Implications

This study is sure to provide the following four important implications for management practice in the post-epidemic era. Firstly, much attention in the enterprises should be paid to the important role of new-generation employees in the organization. The organizational leadership should thoroughly consider their management needs and development platforms of combining rigidity with softness and stimulate their potentials for performance improvement to be transformed into their performance action strength, thus contributing to whatever they can do for the enterprise sustainable development in the post-epidemic era.

Then, due to the fact that the paradoxical leadership can promote new-generation employees' adaptive performance, the organization should pay more attention to the management leadership training in the team with higher requirements for the adaptive behavior of new-generation employees. Training and learning enable managers to appropriately adopt the paradoxical leadership mode in order to be capable of guiding employees to deal with any unpredictable environment or working tasks in a reasonable way, so as to purposefully to achieve the results of the improvement of the employee adaptive performances.

Furthermore, the important role of harmonious work passion reminds organization managers to pay enough attention to the triggering of harmonious work passion for new-generation employees. New-generation employees should be given enough support in work and care in life, thereby ensuring that they psychologically feel more secure and their sense of organization-identity can be improved. Consequently, the harmonious work passion of new-generation employees can be purposefully stimulated. Similarly, if new-generation employees can be also be appropriately given a free working space, their enthusiasm and initiative to participate in the organizational work can be greatly increased, with more harmonious work passion to be aroused.

Finally, leaders in the organization should also pay sufficient attention to the individual differences of new-generation employees, and implement their management with specific pertinence. As for those employees who possess a high sense of core self-evaluation, more paradoxical leadership methods can be applied, and more support as well as more challenges can be given in order to promote their harmonious work passion, thus promoting their adaptive performance. However, those who are characterized with low core self-evaluation would be better treated with more self-identity and compliments/praise to enhance their self-value and ability assessment. In another way, the core self-evaluation management can also be integrated into the organizational cultural activities and enabled to cultivate employees' sense of core self-evaluation through course-training programs or team-building activities. The overall core self-evaluation level can also be highlighted by upgrading the selection standards during employee recruitment.

### 6.3. Research Limitations and Prospects

The research limitations and prospects are mainly manifested in the following four aspects. To begin with, in this study, the data collection was proceeded via the employee self-evaluation and employee evaluation leadership, so the objectivity of data acquisition needs to be further enhanced. The deviation, due to the common methods, could be unavoidable because of some subjectivities during the employee self-evaluation process.

Even if the common method bias is confirmed to be not obvious, multi-source data should be utilized in future research and the method of data collection should also be optimized.

Second, as for the inspective examination of the boundary conditions of the paradoxical leadership influence effect, the role of the core self-evaluation element on the paradoxical leadership based on the self-determination theory was well examined in this paper. Factors such as the employees' paradoxical mentality, the employees' recognition for their leaders and the power distance between the leaders and the employees may also affect the effect of paradoxical leadership, which will be regarded as necessary perspectives for future research so that paradoxical leadership research can be further promoted.

Thirdly, the study samples in this paper were only collected from some areas in China. The current sample size is thought be relatively small. In order to explore whether the situation is similar in different regions of other countries, it is necessary to select more samples from more different regions of different countries so as to increase the sample size in the future.

Finally, although this study adopted the method of collecting questionnaires at two time points, the questionnaires of paradoxical leadership and harmonious work passion were still collected at the same time. There are some defects in the analysis of causality by cross-sectional data. In the later work, longitudinal research can be used, such as tracking employees in multiple countries and regions at different time periods, to further clarify the causal relationship between variables.

## 7. Conclusions

In the previous studies on paradoxical leadership, the positive or negative influences of paradoxical leadership have been confirmed and emphasized by the majority of scholars, but there is a lack of in-depth discussion on the relationship between paradoxical leadership and adaptive performance of new-generation employees and the influence mechanism. Based on the self-determination theory, this paper conducted research into the relationship between paradoxical leadership and adaptive performance of new-generation employees via the questionnaire survey. The empirical results generally supported the hypothesis proposed in this paper. Specifically, we found that: (1) Paradoxical leadership is positively correlated with adaptive performances of new-generation employees, which indicates that paradoxical leadership can promote new-generation employees to produce more adaptive behaviors and achieve higher adaptive performance. (2) Harmonious work passion has a mediating role in the relationship between paradoxical leadership and adaptive performance of new-generation employees, which indicates that paradoxical leadership using dialectical and unified thinking and behavior can better stimulate harmonious work passion of new-generation employees, thus bringing about a high-level adaptive performance. (3) Core self-evaluation positively moderates the relationship between paradoxical leadership and harmonious work passion and moderates the mediating effect of harmonious work passion between paradoxical leadership and adaptive performance of new-generation employees, which indicates that core self-evaluation has a positive role in the work situation.

**Author Contributions:** N.L. is responsible for the study design and the manuscript proofreading. M.D. is responsible for the survey and the manuscript writing. All authors have read and agreed to the published version of the manuscript.

**Funding:** This study was supported by the National Natural Science Foundation of China (No. 52174184).

**Institutional Review Board Statement:** Ethical review and approval were waived for this study because the experimental part of this article was only a questionnaire survey and did not involve human trials, and the consent of the subjects was obtained during the questionnaire survey.

**Informed Consent Statement:** Informed consent has been obtained from all participants in the study.

**Data Availability Statement:** Data are made available by authors according to reasonable requirements.

**Acknowledgments:** The authors acknowledge all the participants in the study.

**Conflicts of Interest:** The authors declare no potential conflict of interest.

## Appendix A

**Table A1.** Measurement items.

| Variables | Items |
|---|---|
| Paradoxical Leadership | 1. He/She uses a fair approach to treat all subordinates uniformly, but also treats them as individuals.<br>2. He/She puts all subordinates on an equal footing, but considers their individual traits or personalities.<br>3. He/She communicates with subordinates uniformly without discrimination, but varies communication styles depending on their individual characteristics or needs.<br>4. He/She manages subordinates uniformly, but considers individualized needs.<br>5. He/She assigns equal workloads, but considers individual strengths and capabilities to handle different tasks.<br>6. He/She shows a desire to lead, but allows others to share the leadership role.<br>7. He/She likes to be the center of attention, but allows others to share the spotlight as well.<br>8. He/She insists on getting respect, but also shows respect toward others.<br>9. He/She has a high self-opinion, but shows awareness of personal imperfection and the value of other people.<br>10. He/She is confident regarding personal ideas and beliefs, but acknowledges that he/she can learn from others.<br>11. He/She controls important work issues, but lets subordinates handle details.<br>12. He/She makes final decisions for subordinates, but lets subordinates control specific work processes.<br>13. He/She makes decisions about big issues, but delegates lesser issues to subordinates.<br>14. He/She maintains overall control, but gives subordinates appropriate autonomy.<br>15. He/She stresses conformity in task performance, but allows for exceptions.<br>16. He/She clarifies work requirements, but does not micro-manage work.<br>17. He/She is highly demanding regarding work performance, but is not hypercritical.<br>18. He/She has high requirements, but allows subordinates to make mistakes.<br>19. He/She recognizes the distinction between supervisors and subordinates, but does not act superior in the leadership role.<br>20. He/She keeps distance from subordinates, but does not remain aloof.<br>21. He/She maintains position differences, but upholds subordinates' dignity.<br>22. He/She maintains distance from subordinates at work, but is also amiable toward them. |
| Harmonious Work Passion | 23. This work allows me to live a variety of experiences.<br>24. The new things that I discover with this work allow me to appreciate it even more.<br>25. This work allows me to live memorable experiences.<br>26. This work reflects the qualities I like about myself.<br>27. This work is in harmony with the other activities in my life.<br>28. For me, it is a passion that I still manage to control.<br>29. I am completely taken with this work. |
| Adaptive performance | 30. Adapted well to changes in core tasks.<br>31. Coped with changes to the way I have to do my core tasks.<br>32. Learned new skills to help me adapt to changes in my core tasks.<br>33. Dealt effectively with changes affecting my work unit (e.g., new members).<br>34. Learnt new skills or taken on new roles to cope with changes in the way my unit works.<br>35. Responded constructively to changes in the way my team works.<br>36. Responded flexibly to overall changes in the organization (e.g., changes in management)<br>37. Coped with changes in the way the organization operates.<br>38. Learnt skills or acquired information that helped me adjust to overall changes in the organization. |
| Core Self-Evaluations | 39. I am confident I get the success I deserve in life.<br>40. Sometimes I feel depressed.<br>41. When I try, I generally succeed.<br>42. Sometimes when I fail I feel worthless.<br>43. I complete tasks successfully.<br>44. Sometimes, I do not feel in control of my work.<br>45. Overall, I am satisfied with myself.<br>46. I am filled with doubts about my competence. |

**Table A1.** *Cont.*

| Variables | Items |
|---|---|
| Core Self-Evaluations | 47. I determine what will happen in my life.<br>48. I do not feel in control of my success in my career.<br>49. I am capable of coping with most of my problems.<br>50. There are times when things look pretty bleak and hopeless to me. |

Note: Paradoxical Leadership (22-item scale developed by Zhang, Y.; Waldman, D.A.; Han, Y.L. & Li, X.B. 2015 [15]); Harmonious Work Passion (7-item scale developed by Vallerand, R.J.; Blanchard, C.; Mageau, G.A.; Koestner, R.; Ratelle, C.; Leonard, M.; Gagne, M. & Marsolais, J. 2003 [51]); Adaptive performance (9-item scale developed by Griffin, M.A.; Neal, A. & Parker, S.K. 2007 [3]); The Core Self Evaluations (12-item scale developed by Judge, T.A.; Erez, A.; Bono, J.E. & Thoresen, C.J. 2003 [31]).

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
