# Peer review of "The Influence of Paradoxical Leadership on Adaptive Performance of New-Generation Employees in the Post-Pandemic Era: The Role of Harmonious Work Passion and Core Self-Evaluation"

_sustainability, doi:10.3390/su142114647_

Round 1

Reviewer 1 Report

Dear Authors of the manuscript "The Influence of Paradoxical Leadership on Adaptive Performance of New-generation Employees in the Post-pandemic Era: The Role of Harmonious Work Passion and Core Self-evaluation".

Please see below my recommendations for the improvement of your proposal.

The Introduction section is well supported by previous researches from literature, but you should organize the things in a better manner.

At the end of the Introduction, I recommend you to define in 2-3 short paragraphs the following issues: the research gap, the research goal, the research questions, the objectives of the article.

At this moment, these issues are described within the Introduction, but in a diluted manner.

If you summarize them at the end of the Introduction, the reader will understand your approach in a better way.

Figure 1 appears too early in the article, long before the justification from the Literature Review chapter.

Normally, figure 1 should be the result of the literature review, so I recommend you to place it at the end of this chapter.

More than this, you should improve it by drawing the symbols H1, H2, H3, H4 on the arrows.

The Literature Review section should be enriched with the following references from the extant literature: https://doi.org/10.3390/ijerph18073505, https://doi.org/10.1525/cmr.2014.56.3.58, https://doi.org/10.3390/electronics11162535, https://doi.org/10.1057/s41291-018-0043-9, https://doi.org/10.3390/admsci10040096, https://doi.org/10.3390/ijerph17186724.

By including these references, your context will be more clear for the readers.

At rows 457-458 you have a sentence that must be revised: "In order to improve the enthusiasm of the participants and their willingness with which to answer the match-question survey."

I don't understand the sentence; the predicate is missing. Please revise and correct it.

Also, the sentence from rows 463-466 must be revised.

Between rows 475-492 you discuss about the questions included in the survey. I recommend you to add an appendix at the end of the article and present a table with the questions, so that the reader can see them in a synthetic image.

In the "5.3. Research Limitations and Prospects" section, I recommend you to also take into consideration another future research direction: a longitudinal study, based on many periods of time and many regions/countries.

Best Regards!

Reviewer 2 Report

This paper presents a very interesting study.  The methodology is well designed and justified.  Congratulations for the well-conducted study.  Having said this, I believe your study has a number of significant contributions to the field of sustainability.  However, you have not highlighted the contributions enough in the Discussion of the Findings section.  The Discussion section is where you present you findings, point out how they are similar to or different from previous studies, and highlight your contributions to the field.   The Discussion section should not contain implications of the findings.  I suggest you separate the implications from the discussions by making it a section of its own before the Conclusions section.

In highlighting the contributions of your study, you may consult the literature on the theory of corporate sustainability and organizational theory of resilience since both contains the paradox perspective and the perspective of organizational adaptive capacity. 

Round 2

Reviewer 1 Report

Dear Authors,

The revised version of the article addressed all my concerns. Now I have a minor recommendation: under figure 1 and figure 2, please specify the source: "Own concept" or "Authors' proposal".

Best Regards!
